# Antibacterial Fractions from *Erodium cicutarium* Exposed—Clinical Strains of *Staphylococcus aureus* in Focus

**DOI:** 10.3390/antibiotics11040492

**Published:** 2022-04-06

**Authors:** Vanja Ljoljić Bilić, Uroš M. Gašić, Dušanka Milojković-Opsenica, Hrvoje Rimac, Jadranka Vuković Rodriguez, Josipa Vlainić, Diana Brlek-Gorski, Ivan Kosalec

**Affiliations:** 1Faculty of Pharmacy and Biochemistry, University of Zagreb, 10000 Zagreb, Croatia; vljoljic@pharma.hr (V.L.B.); hrimac@pharma.hr (H.R.); 2Institute for Biological Research “Siniša Stanković”—National Institute of Republic of Serbia, University of Belgrade, Bulevar Despota Stefana 142, 11060 Belgrade, Serbia; uros.gasic@ibiss.bg.ac.rs; 3Faculty of Chemistry, University of Belgrade, Studentski trg 12-16, 11158 Belgrade, Serbia; dusankam@chem.bg.ac.rs; 4Internationale Lerchen Apotheke, 80809 Munich, Germany; nadadedada@gmail.com; 5Laboratory for Advanced Genomics, Division of Molecular Medicine, Rudjer Bošković Institute, 10000 Zagreb, Croatia; josipa.vlainic@irb.hr; 6Croatian Institute of Public Health, Rockefeller Str. 7, 10000 Zagreb, Croatia; diana.brlek-gorski@hzjz.hr

**Keywords:** *Erodium cicutarium*, MRSA, biofilm, bioautography, fractionation, anti-hemolytic, phenolic composition, galloyl-shikimic acid

## Abstract

Followed by a buildup of its phytochemical profile, *Erodium cicutarium* is being subjected to antimicrobial investigation guided with its ethnobotanical use. The results of performed in vitro screening on *Staphylococcus aureus*, *Pseudomonas aeruginosa*, and *Candida albicans* strains, show that *E. cicutarium* has antimicrobial activity, with a particular emphasis on clinical *S. aureus* strains—both the methicillin sensitive (MSSA) and the methicillin resistant (MRSA) *S. aureus*. Experimental design consisted of general methods (the serial microdilution broth assay and the agar well diffusion assay), as well as observing bactericidal/bacteriostatic activity through time (the “time-kill” assay), investigating the effect on cell wall integrity and biofilm formation, and modulation of bacterial hemolysis. Observed antibacterial activity from above-described methods led to further activity-guided fractionation of water and methanol extracts using bioautography coupled with UHPLC-LTQ OrbiTrap MS^4^. It was determined that active fractions are predominantly formed by gallic acid derivatives and flavonol glycosides. Among the most active phytochemicals, galloyl-shikimic acid was identified as the most abundant compound. These results point to a direct connection between galloyl-shikimic acid and the observed *E. cicutarium* antibacterial activity, and open several new research approaches for future investigation.

## 1. Introduction

Antimicrobial activity of naturally occurring compounds is constantly being investigated and a need for new approaches is higher than ever, particularly due to a global increase in antimicrobial resistance [1,2]. This work was designed in order to explore the, so far, poorly investigated antimicrobial activity of *Erodium cicutarium* (L.) L’Hér. ex Aiton (Geraniaceae) plant extracts. *Erodium cicutarium* is a native species to the Mediterranean, but wide spread over the world, with fern-like, pinnate leaves forming a rosette, and small pink flowers, which after flowering transform to fruits with mericarps joined together in a spine-like style [3,4,5,6]. Literature data, as well as our previous research, show the presence of a rich polyphenolic composition, including tannins, flavonoids and hydroxycinnamic acids, as well as saponins, fatty acids, sugars and amino acids, along with a chemically diverse essential oil (hexadecanoic acid being the standing out compound) [6,7,8,9,10,11,12,13,14].

Literature data, as well as our previous work, show a wide diversity of the phytochemical composition of *E. cicutarium* extracts, which motivated our bioactivity research [6,7,10,11,12,13,14,15,16]. According to Munekata et al., the *Erodium* genera, particularly species *E. absinthoides*, *E. cicutarium* and *E. glaucophyllum*, should be considered as potential sources of natural antimicrobial compounds [12]. This paper presents an overview of in vitro antimicrobial activity results, starting with the phase of the whole and complex, but extensively profiled plant extracts and concluding with isolated and, again, extensively phytochemically characterized antimicrobial fractions. The obtained results present an opportunity for deliberation about the contribution of each found compound in the final antimicrobial effect, as well as their potential synergistic interactions. Present, as well as relatively scarce literature data, describe *E. cicutarium* antimicrobial activity, but except for classical screenings, e.g., the diffusion and (micro)dilution methods, no detailed or more in depth studies were presented [9,10,12,17].

To the best of our knowledge, no data on time-dependence of the *E. cicutarium* activity, the anti-biofilm activity, or overall analysis of separated active segments of the complex *E. cicutarium* extracts is available. Thus, the aim of the present work is to take one step forward towards improving the scientific knowledge on *E. cicutarium* and its antimicrobial potential. Since a few studies demonstrated a direct relation, elucidating the main compounds associated with the observed antibacterial activity is of particular interest. This especially pertains to the obtained activity data on *Staphylococcus aureus* strains, including the methicillin resistant strains (MRSA), and potentially opens new doors for further in-depth antimicrobial activity research in the future.

## 2. Results and Discussion

### 2.1. Serial Microdilution Broth Assay and Agar Well Diffusion Assay

The agar well diffusion assay and the serial microdilution broth assay were used for an initial antimicrobial activity screening of *E. cicutarium* extracts and present an overview of their activity towards Gram-positive (*S. aureus*) and Gram-negative (*P. aeruginosa*) bacterial experimental models, as well as a yeast representative (*C. albicans*). The results showed activity on the *S. aureus* model, which led to a further tests including clinical isolates of methicillin sensitive (MSSA) and methicillin resistant (MRSA) strains (Table 1).

A total of eight *E. cicutarium* extract samples (two types of extracts—water and methanolic, from four localities in Croatia (Podvinje, Plitvice, Trešnjevka, Buzin) were tested and compared (Table 2). Statistical comparison was used to compare activity between the samples from different localities of the same extract type (*p* < 0.05, one-way ANOVA and Tukey’s post hoc test), as well as between different extracts type for the same location (*p* < 0.05; *t*-test).

### 2.2. “Time-Kill” Assay

The sample from the Podvinje locality stands out as the most active in the agar well diffusion assay. Podvinje-W (Podvinje water extract) was the most active extract against *S. aureus* ATCC 6538 and 10663, as well as the *P. aeruginosa* ATCC 27853 model. Zones of growth inhibition (ZI) are in a range from 16 ± 1 mm (for *P. aeruginosa* ATCC 27853) to 22 ± 1 mm (for MSSA 10663). Podvinje-M (Podvinje methanolic extract) showed the highest activity against MRSA MFBF 10679, MSSA MFBF 10663 and *C. albicans* ATCC 90028. ZI range is from 15 ± 2 mm (*C. albicans* ATCC 90028) to 22 ± 1 mm (MSSA MFBF 10663). The Plitvice-M (Plitvice methanol extract) sample showed a similar activity as the Podvinje-M sample in the case of MRSA MFBF 10679 (ZI = 19 ± 1 mm) and MSSA MFBF 10663 (ZI = 22 ± 1 mm). Similarly, in the serial microdilution broth assay, Podvinje samples were again the most active samples. Podvinje-M was the most active sample in case of *S. aureus* ATCC 6538, MRSA MFBF 10679 and 10663, with minimal inhibitory concentration (MIC) values from 2.50 ± 0.00 mg/mL (MRSA MFBF 10679) to 3.75 ± 2.17 mg/mL (*S. aureus* ATCC 6538). The lowest MIC value for Podvinje-W was against MSSA MFBF 10663 (3.33 ± 1.44 mg/mL). Trešnjevka-M also showed a relatively high activity, with MIC ranges of 2.50 ± 0.00 mg/mL (for MRSA MFBF 10679), 3.33 ± 1.4 mg/mL (for MSSA MFBF 10663), and 3.75 ± 2.17 mg/mL (for *S. aureus* ATCC 6538).

A statistically significant difference in antimicrobial activity among the water and methanolic extract types from the same location was present in Trešnjevka and Buzin localities (*p* < 0.05; *t*-test). Methanolic samples were more active, with the exception of Buzin-W effect on MSSA MFBF 10663. In the case of *P. aeruginosa* ATCC 27853, the most active sample was Buzin-M (MIC 10.00 ± 0.00 mg/mL). Experiments also showed that in the case of *C. albicans* ATCC 90028, MIC values were not detected in the tested concentration range ≤ 20 mg/mL. In the agar well diffusion assay activity was present only for methanolic extracts, with water extracts showing no activity (c = 20 mg/mL).

After the initial antimicrobial activity results for the microbial strains tested, samples from the Podvinje locality were chosen for in-depth antimicrobial examination of E. cicutarium extracts.

A search through the existing literature demonstrates that antimicrobial activity data of *E. cicutarium* extracts are relatively scarce. According to Bussmann et al. [17], MIC of *E. cicutarium* water extract on *S. aureus* ATCC 25923 is 4 mg/mL, and is similar to the results obtained in this work. In the same work, MIC for the ethanolic extract was shown to be 16 times higher (64 mg/mL), while our data show that the alcoholic, i.e., methanolic extracts, were in general more active on the investigated microbial models. Bussmann et al. [17] chose *Escherichia coli* ATCC 25922 as the Gram-negative bacterial model with a MIC of 16 mg/mL. In this work, *P. aeruginosa* ATCC 27853 was chosen as the Gram-negative model, following the traditional use of *E. cicutarium* in skin and tissue infection treatments, and the lowest observed MIC value of 10.0 ± 0.00 mg/mL was achieved by the Buzin-M sample.

In the work of Nikitina et al. [10], *E. cicutarium* water and ethanolic extracts showed bacteriostatic effects on microbial species naturally occurring in soil, e.g., *Bacillus* sp., *Azotobacter* sp., and *Pseudomonas* sp. It was shown that the water extract was more active than the ethanolic extract, when tested in same 1 mg/mL concentrations. The authors’ explanation for these results is the fact that a higher amount of polar phenolic compounds, mainly polyphenols, is present in the water extract. Also, polyphenols and their oxidation products have the ability to inhibit enzymes and could be the reason for the observed results, correlating with bacteriostatic activity of plant phenolic compounds of the families Geraniaceae and Rosaceae and their antioxidant potential [10]. Since the applied methodology did not follow current guidelines, e.g., EUCAST (European Committee on Antimicrobial Susceptibility Testing) or CLSI (Clinical and Laboratory Standards Institute), a direct comparison to our results is not possible.

Gohar et al. [18] investigated antibacterial activity of geraniin from *Erodium glaucophyllum* on *S. aureus, E. coli,* and *C. albicans*; MICs were 3.16, 2.5 and 1.99 mg/mL, respectively.

Bouaziz et al. [12] tested *E. cicutarium* methanolic and ethyl-acetate extracts on several bacterial species, including *S. aureus*, *P. aeruginosa*, *E. coli*, *Salmonella enterica,* and *Bacillus subtilis* (ZI was 10–11 mm), while antifungal activity on *C. albicans* could not be confirmed.

The *Erodium* genera is known for its species with high essential oil content, and *E. cicutarium* is not an exception [8,9,19]. Although plant extracts are the main subject of investigation in this work, the indirect contribution of the present essential oil and its constituents towards the antimicrobial activity of extracts should also be included in the general deliberation of observed results. Antimicrobial activity of the essential oil was previously shown by Stojanović-Radić et al. [9] on several microbial species, including bacterial and fungal species, e.g., *S. aureus*, *P. aeruginosa*, *E. coli*, and *C. albicans*. *E. cicutarium* essential oil analysis was the subject of our previous work as well, with a total yield between 0.03% and 0.09%, where hexadecenoic acid was found to be the major component (41.5–49.6%) [6]. This finding is in accordance with other literature data, such as the work by Radulović et al. [8].

Different extract types show different antimicrobial activity on *S. aureus* experimental models in relation to the activity time frame and bacterial viability, i.e., resulting in bacteriostatic or bactericidal activity. For the Podvinje-W sample, the activity onset is t_9_-t_24_ h, and is bacteriostatic for MSSA MFBF 10663 and bactericidal after 24 h for MRSA MFBF 10679 (Figure 1). Compared to Podvinje-W, the Podvinje-M sample has an earlier bactericidal activity onset, visible after only 3 h for both bacterial strains (Figure 1). This difference in activity can be due to a difference in phytochemical composition, including the types of phytochemicals present, as well as their amount in each extract (see Section 2.5. TLC and bioautography for further discussion).

When visualizing bacterial viability during the time under the effect of investigated extracts, a statistically significant difference can be observed already after 1 h (t_1_) (*p* ≤ 0.01; one-way ANOVA and Tukey’s post hoc test), but is more prominent after > 3 h (*p* ≤ 0.0001; one-way ANOVA and Tukey’s post hoc test) (Figure 2).

### 2.3. Modulation of Cell Wall Integrity

Due to a relatively fast manifestation of bactericidal activity against *S. aureus* in the case of methanolic extracts, the question of possible mechanism of action was directed towards the cell wall and/or membrane integrity disruption/modulation. Even with a 5× MIC concentration of the Podvinje-M extract tested in a broad time frame (at 0, 1, 2, 3, 18 and 24 h), no protein leakage was detected from treated bacterial cells at 280 nm (Figure 3).

Conversely, a visible leakage was present when a positive control (Triton X-100; 10%, m/V) was used; with values of 1521.7 ± 173.3 µg/mL in the first hour of measurement, and reaching the maximal quantity after 3 h (1970.0 ± 70.0 µg/mL), as was expected for a surfactant (one-way ANOVA and Tukey’s post hoc test; **, *p* ≤ 0.01; ***, *p* ≤ 0.001; ****, *p* ≤ 0.0001). An absence of released cellular proteins is also visible in the case of a negative control (untreated bacteria).

### 2.4. Evaluation of Anti–Biofilm Activity

Biofilm formation of *S. aureus* (MSSA) ATCC 6538 and MRSA MFBF 10679 was inhibited by both the Podvinje-W and the Podvinje-M extracts. The MBFIC_50_ and MBFIC_90_ values represent the lowest extract dilutions at which bacterial biofilm mass was inhibited during formation by 50% and 90%, respectively, compared to a negative control (inocula with broth). MBFICs were calculated via linear regression of log_10_ (concentration of extracts) vs. % biofilm reduction. In general, the MBFIC_50_ and MBFIC_90_ are lower for MSSA ATCC 6538 than for MRSA MFBF 10679 (Table 3). For MSSA ATCC 6538, the MICSB_50_ and MICSB_90_ for both extracts are approximately two-fold lower for the Podvinje-M sample (0.41 ± 0.27 µg/mL and 3083.98 ± 549.69 µg/mL, respectively), than for the Podvinje-W sample (1.05 ± 0.97 µg/mL and 5095.37 ± 1143.88 µg/mL). In the case of the MRSA MFBF 10679, again, the methanolic extract had a higher anti-biofilm activity than the water extract. Values for Podvinje-M in this case were MICSB_50_ = 3.17 ± 5.30 µg/mL and MICSB_90_ = 8091.00 ± 477.24 µg/mL, again higher compared to the MSSA ATCC 6538 strain.

### 2.5. Modulation of Bacterial Hemolysis

Subinhibitory concentrations (*c*_1_ = MIC/2; *c*_2_ = MIC/4) of both extracts from the Podvinje locality reduce hemolytic activity of the tested *S. aureus* strains (MSSA and MRSA), with higher concentrations exhibiting higher anti-hemolytic activity. Although statistically not significant, it can be seen that the water extract has a higher inhibitory activity than the methanolic extract (Figure 4).

The Podvinje-W sample at concentration *c*_1_ was the most active sample, inhibiting hemolytic activity of both strains in a similar amount, about 50%. A statistically significant difference of anti-hemolytic activity is evident between *c*_1_ and *c*_2_ of Podvinje-W on MRSA MFBF 10679 (*t*-test, *p* = 0.0029), being 48.7% and 19.6%, respectively, (Figure 4). There was no statistically significant difference between the activity among strains (MSSA and MRSA).

The degree of hemolysis was also calculated and expressed as bovine hemoglobin equivalents (mg/mL) (Table 4). Acceptable experimental growth conditions regarding produced hemolysins are apparent when comparing the hemoglobin concentration released from erythrocytes caused by nontreated bacteria (7.53–8.42 mg/mL) and a positive control treated with Triton X-100 (2%, m/V) (8.39 mg/mL). In the case of the most active sample, Podvinje-W (*c*_1_), hemolysis was inhibited to a hemoglobin concentration of 3.47 mg/mL for *S. aureus* ATCC 6538 and 3.90 mg/mL for MRSA MFBF 10679 (Table 4).

### 2.6. TLC and Bioautography In Situ

After performing TLC chromatogram development of Podvinje-W and Podvinje-M extracts (Figure 5), their bioautography with *S. aureus* ATCC 6538 and semi-preparative isolation of the observed active fractions, UHPLC-LTQ OrbiTrap MS^4^ identification was performed (Table 5).

The Podvinje-W sample TLC chromatogram revealed six zones after visualization with NSR (1%, m/V) at 366 nm, with retardation factor values (*R*_F_) as follows, *R*_F1_ = 0.62, *R*_F2_ =0.66, *R*_F3_ = 0.69, *R*_F4_ = 0.72, *R*_F5_ = 0.75, and *R*_F6_ = 0.77. The Podvinje-M chromatogram also showed six separated zones, with *R*_F_s as follows, *R*_F1_ = 0.67, *R*_F2_ = 0.71, *R*_F3_ = 0.74, *R*_F4_ = 0.77, *R*_F5_ = 0.79, and *R*_F6_ = 0.81

The phytochemical composition of these zones was analyzed via UHPLC-LTQ OrbiTrap MS^4^ and showed that galloyl-shikimic acid is the most abundant compound of the antimicrobially active zones of both extracts (Table 5). A total of 27 compounds were identified in the isolated active fraction of the Podvinje-M sample (the extract that showed bactericidal activity in the time-kill assay) and a total of 24 compounds were found in the active fraction of the Podvinje-W sample (the sample that showed bacteriostatic activity in the time-kill assay). The identified compounds can be generally classified into two groups, gallic acid derivatives and flavonol glycosides. Gallic acid derivatives (compounds **1**–**19** in Table 5) were quantified as mg of gallic acid equivalents (GAE) per kg of fraction. The flavonol glycosides group (compounds **20**–**27** in Table 5) were quantified as mg of rutin equivalents (RE) per kg of fraction. Compounds under numbers **3**, **6**, **15**, **16**, **17**, **19**, and **26** in the active fractions (Table 5) were reported in our previous work as compounds detected for the first time, not only in *E. cicutarium*, but also in the Geraniaceae family [14]. There was one flavonol glycoside among them, isorhamnetin 3-*O*-(6′-rhamnosyl)glucoside (narcissin), and also, several gallic acid derivatives: galloyl pentoside isomer 1 and isomer 2, methylgalloyl-caffeoyl hexoside, methylgalloyl-coumaroyl hexoside, trimethylellagic acid isomer 1 and isomer 2. When comparing the two samples, the three missing compounds in the Podvinje-W sample were ellagic acid, galloyl hexoside isomer 2, and methylgalloyl hexoside isomer 2; all of these were gallic acid derivatives (Table 5).

Their quantities in the Podvinje-M fraction were 11.77 mg/kg GAE, 9.02 mg/kg GAE and 15.05 mg/kg GAE, respectively. The galloyl-shikimic acid content in the antimicrobially active fractions in the Podvinje-M sample was 375.42 mg/kg GAE, lower than in the Podvinje-W sample (531.45 mg/kg GAE), which could potentially lead to a conclusion that the observed bactericidal activity of the Podvinje-M sample is a result of a synergistic activity of several compounds in the extract, as well as of their amount. This might indicate the need to deliberate on gallic acid derivatives as the most responsible compounds for the antimicrobial activity of *E. cicutarium* extracts. Gallic acid content was about 2.5-fold higher in the Podvinje-M sample (140.51 mg/kg GAE), than in the Podvinje-W sample (54.64 mg/kg GAE). The protocatechuic acid and the methylgallate content were approximately 6-fold higher in the Podvinje-M sample (22.60 mg/kg GAE and 24.67 mg/kg GAE, respectively), than in the Podvinje-W sample (3.82 mg/kg GAE and 4.10 mg/kg GAE, respectively). Since gallic acid derivatives were detected in the highest amount in the active fractions, their biologic significance may be discussed. Gallic acid is known to be present in the *Erodium* genera, and has diverse bioactive properties, including antimicrobial activity against human pathogens [7,20]. Its antimicrobial action is based on several mechanisms of action, including changes of membrane potential, disruption of cell membrane, inhibition of bacterial proliferation and formation of biofilms, as well as increase in susceptibility of resistant strains towards β-lactams (e.g., MRSA) [21,22,23,24,25]. In this study, the potential mechanism of action was not thoroughly tested, but no cell-wall disruption was observed (Figure 3). Ellagic acid, as a naturally occurring polyphenol component, can be present in a free form, or in more complex forms of ellagitannins and glucosides. Similar to gallic acid, it is known to possess various activities, also including in vitro antibacterial activity (e.g., on *Helicobacter pylori*) [26]. Ellagitannins are a diverse group of esters, which belong to a class of hydrolysable tannins and, as such, are slowly hydrolyzed in the digestive tract, releasing ellagic acid [27]. Classified as nutraceuticals, their biological activity is diverse, and among others, they exhibit antifungal, antiviral and antibacterial activity (including antibiotic-resistant strains, such as MRSA) [27,28]. The antimicrobial effect of tannins is not only influenced by their well-known ability to precipitate proteins, but also seems to be species-specific and influenced by their structure [28]. It is also reported that, similarly to our results, *S. aureus* shows a higher susceptibility to tannins than other bacterial species (e.g., compared to *Clostridiales perfringens, E. coli*, and *Lactobacillus plantarum*) [28]. As for their proposed mechanisms of action, they are generally considered to bind to adhesins, to inhibit extracellular enzymes and oxidative phosphorylation, as well as to participate in the disruption of cellular membrane permeability, substrate deprivation, and to form complexes with cell wall and metal ions [28,29]. Regarding their structure-activity relationship, reports consider the pyrogallol group, as well as the number of free galloyl groups on the glucopyranose cores, to have an important role in their activity (molecules with more pyrogallol groups being more active than those with less groups) [28]. Corilagin, often used as a model for ellagitannins in the literature, exhibits antimicrobial activity against *S. aureus* by inhibiting protein expression and bacterial growth of *E. coli* by disrupting the cell membrane permeability [30]. According to a more recent study by Puljula et al. [28], other ellagitannins showed much higher antimicrobial activity than corilagin and indicate that the antimicrobial effects of ellagitannins, in general, could be more significant than previously thought. According to Cowan, the well-established antimicrobial activity of flavones, flavonoids, and flavonols is caused by their capability to complex extracellular and soluble proteins, as well as cell walls of bacteria, and to disrupt microbial membranes in case of more lipophilic flavonoids [29].

Overall, the extensive phenolic profile of the isolated *E. cicutarium* active fractions presents two groups, gallic acid derivatives and flavonol glycosides, as the most responsible phytochemicals for their in vitro antimicrobial activity, with galloyl-shikimic acid as the most abundant compound. Considering that a few previous studies elucidated such a direct relationship between *E. cicutarium* (and *Erodium* species, in general) with phytochemicals responsible for their antimicrobial activity, the obtained results (especially the anti-*S. aureus* activity, i.e., MRSA strains), show potential for further research.

## 3. Materials and Methods

### 3.1. Plant Material and Sample Preparation

Details on origin and *E. cicutarium* plant material collection data from four localities in Croatia (Podvinje, Plitvice, Trešnjevka, Buzin), along with their voucher specimen numbers and extract preparation method, were described previously [6,14]. Obtained data showed that the phenolic composition was similar among localities, and only in the methanolic extract from Trešnjevka slightly less phenolics were detected [14]. Antimicrobial experiments included a total of eight *E. cicutarium* extracts samples. As described previously [14], the extract preparation included ultrasonication of 2.5 g of powdered herbal material with two types of solvents (12.5 mL)—methanol or water, at 45 °C for 45 min. After the first extraction and filtration, two re-extractions of the same plant material were made, and the combined filtrates were evaporated and freeze-dried [14]. The antimicrobial activity screening was performed on methanol and water extracts of *E. cicutarium* aerial parts from four above-mentioned localities in Croatia. In our previous work, these extracts were extensively phytochemically characterized, focusing on their qualitative and quantitative phenolic profile [14]. In this work, after observing their activity in the screening performed on all samples, the extracts from the Podvinje locality were chosen for further, more detailed activity testing.

### 3.2. Microbial Species, Media and Positive Controls

Antimicrobial experiments were performed with standard laboratory strains: a Gram-positive strain—*Staphylococcus aureus* ATCC (*American Type Culture Collection*) 6538, a Gram-negative strain—*Pseudomonas aeruginosa* ATCC 27853, and a yeast model—*Candida albicans* ATCC 90028, all from *Collection of Microorganisms* stock-cultures of the Department of Microbiology, Faculty of Pharmacy and Biochemistry University of Zagreb (MFBF). Additionally, clinical *S. aureus* strains were included in the experiments—a methicillin sensitive (MSSA MFBF 10663) and a methicillin resistant strain (MRSA MFBF 10679). All microbial media were purchased from Merck (Darmstadt, 64297, Germany). Gentamicin sulphate (Sigma-Aldrich, St. Louis, MO, USA) and nystatin (Pliva, Zagreb, Croatia) were used as a susceptibility quality control of strains and the method.

### 3.3. Antimicrobial Susceptibility Testing of Clinical S. aureus Strains

Antimicrobial susceptibility assays were performed using VITEK^®^ 2 (BioMerieux, Craponne, France), an automated instrument using a turbidimetric method. VITEK^®^ cards for susceptibility testing (AST) were inoculated with 0.5–0.63 McFarland units and incubated according to the instructions for use provided by BioMerieux (AST-P658). The instrument performs its susceptibility analyses by monitoring the growth and activity of the organisms in the wells of the test card. The VITEK^®^ 2 AST cards provide AST results and resistance detection. For detection and differentiation between MRSA and MSSA, the VITEK^®^ 2 AST-P580 test uses a combination of oxacillin and cefoxitin tests to detect mecA/mecC-mediated methicillin resistance. The instrument’s expert system interprets any *S. aureus* isolate that tests positive on cefoxitin screening (MIC > 6 μg/mL on the Vitek 2^®^ system) as oxacillin-resistant [31]. Each new batch number of ID cards was tested with stock culture organisms (*S. aureus* ATCC 6538 and *S. aureus* ATCC 25923). Antimicrobial susceptibility results were expressed as MIC (μg/mL) (Table 1).

### 3.4. Agar Well Diffusion Assay

The agar well diffusion assay, described in *European Pharmacopoeia* [32], was slightly modified. Briefly, inoculum was spread onto the surface of Mueller-Hinton agar for bacteria, and Mueller–Hinton agar with 2% (m/V) glucose for fungi, using sterile swabs. Inocula were prepared using phosphate-buffered saline (PBS; pH 7.4) and fresh cultures of microbial strains. Bacterial species were cultured on tryptic-soy agar (TSA) for 18 h at 37 °C, and fungi on Sabouraud 2% (m/V) glucose agar for 48 h at 35 °C. Inoculum was prepared using physiological saline and adjusted to 0.5 McFarland units (Kisker densitometer, Germany). The final bacteria concentrations were approximately 1.5 × 10^8^ colony forming units, CFU/mL, and fungi 3 × 10^6^ CFU/mL. Agar wells (d = 6 mm) were made using sterile stainless-steel cylinders and filled with 50 μL of each sample solution (c = 20.0 mg/mL). Plates were preincubated at +4 °C for 1 h to enhance diffusion, followed by incubation at +37 °C for 18 h under aerobic conditions in the dark (Sanyo MIR-533, Japan). After incubation, plates were examined by measuring zones of growth inhibition (d, mm) around wells, and antimicrobial activity was evaluated. Gentamicin sulphate (c = 10 µg/mL) and nystatin (c = 1 mg/mL) were used as susceptibility quality control of strains and the method. All tests were performed in quintuplicate, and results were expressed as the mean ± SD (Table 2).

### 3.5. Serial Microdilution Broth Assay

Minimal inhibitory concentrations (MICs) were investigated by the serial microdilution broth assay in Mueller–Hinton broth for bacterial strains, and RPMI with 2% of glucose broth for fungi, according to the European Committee on Antimicrobial Susceptibility Testing (EUCAST) guidelines, with minor modifications [33,34]. Briefly, cell suspensions of bacteria and fungi were freshly prepared and maintained on surface of tryptic-soy agar for 18 h at 35 °C in case of bacterial species, and on Sabouraud 2% (m/V) glucose agar for 48 h at 35 °C for fungi. Inocula were prepared using physiological saline and adjusted using Kisker densitometer (Germany) to 0.5 McFarland units. The final concentrations were approximately 7.5 × 10^6^ CFU/mL for bacteria and 1.5 × 10^5^ CFU/mL for fungi. Serial two-fold microdilution was performed in a concentration range from 0.01 mg/mL to 20.0 mg/mL. MIC was defined as the lowest concentration of investigated compounds that allows no more than 20% of microbial growth compared to the untreated control. After inoculation and incubation during 18 h at 35 °C in the dark, viability of bacterial strains was determined by sub-cultivation of each concentration by transferring 10 µL from each dilution well onto the surface of tryptic-soy agar and re-incubation at the same temperature for 18 h. *Candida albicans* MIC was defined as the lowest concentration of tested compounds which allows no more than 20% growth of yeast cells after 24 h incubation, and re–incubation of 10 µL samples transferred onto the surface of Sabouraud 2% (m/V) glucose agar for 48 h at 35 °C. Gentamicin sulphate (c = 10 µg/mL) and nystatin (c = 1 mg/mL) were used as susceptibility quality control of strains and the method. All tests were performed in triplicate and results were expressed as mean values ± SD (Table 2).

### 3.6. “Time-Kill” Assay

After observing antimicrobial activity in the previously performed screening, *S. aureus* was chosen as the model microorganism for further antimicrobial activity investigation. The time-kill assay was performed on two strains, MSSA MFBF 10663 and MRSA MFBF 10679. *E. cicutarium* water and methanol extracts from the Podvinje locality (Podvinje, Croatia) were dissolved in the Mueller-Hinton broth (MHB) (c = 20 mg/mL). Gentamicin sulphate in MHB (50.0 µg/mL) was used as a positive control. The method by Jakas et al. was used, but with a few modifications [35]. The final bacterial concentration was approximately 1.5 × 10^7^ CFU/mL, and at certain time points (0, 1, 3, 6, 9 and 24 h of incubation), 100 μL aliquots were removed from each culture flask in duplicate and were serially diluted ten-folds in sterile saline (from 10^−1^ to 10^−6^). New 100 μL aliquots of each dilution were then transferred onto TSA plates and incubated for 24 h at 37 °C (aerobically, in the dark) for colony count determination. The last two dilutions with growing bacterial colonies of MSSA and MRSA were counted and the number of living bacterial cells (MSSA and MRSA) for each time point was calculated as mean (CFU/mL). The time–kill curve was plotted as log_10_ CFU/mL of living cells of MSSA and MRSA treated with *E. cicutarium* extracts vs. time (h) and compared to a negative control (non-treated bacterial cells incubated under the same conditions).

### 3.7. Modulation of Cell Wall Integrity

Since *E. cicutarium* methanol extract from the Podvinje locality (Podvinje-M) showed bactericidal activity in the time-kill assay, it was taken for further testing of its potential effect on *S. aureus* cell membrane and cell wall integrity. For this, the method according to Zorić et al. was modified [36]. The release of intracellular proteins in bacterial cell supernatants was measured at 280 nm. Cell suspensions prepared from fresh overnight cultures of *S. aureus* ATCC 6538 in PBS (pH 7.4) contained ~3 × 10^8^ CFU/mL bacterial cells and were treated with the Podvinje-M extract (c = 5 × MIC) in different time intervals (0, 1, 2, 3, 18 and 24 h) under aerobic conditions at 37 °C, while rotating (250 rpm). Triton X-100 (10%; m/V) served as a positive control and untreated bacterial cells served as a negative control. After each treatment period, samples were centrifuged (2 min at 1250 rpm) and the release of intracellular material (proteins) in the supernatants was evaluated spectrophotometrically (Biospec Nano, Shimadzu Corporation, Kyoto, Japan).

### 3.8. Evaluation of Anti–Biofilm Activity

To evaluate the effect of investigated plant extracts (Podvinje-M and Podvinje-W) on *S. aureus* ATCC 6538 and MRSA MFBF 10679 biofilm formation, a crystal violet assay was performed according to Vlainić et al. [37], with a few modifications. The assay was performed in sterile 96-well flat-bottom plastic tissue culture plates (TPP, Trasadingen, Switzerland). An amount of 100 µL of a bacterial suspension (~7.5 × 10^6^ CFU/mL in PBS) was incubated aerobically for 24 h at 37 °C with in a concentration range from 0.04 × MIC to 5 × MIC. After incubation and aspiration, wells were washed using PBS (4×) and shaken, followed by fixation of remaining bacterial cells with methanol (20 min). Next day, after drying, the attached biofilm mass was stained with crystal violet (0.5% W/V; 10 min). The plates were rinsed under tap water to remove the rest of the stains and were left to dry. Acetic acid (33%, V/V) was used to resolubilize the stains from adherent cells. Optical density of each well was measured at 540 nm using microtiter plate reader (Azure Ao Absorbance Microplate Reader, Agilent technologies, Vermont, VT, USA). Gentamicin sulphate was used as a positive control (concentration range 0.16–20.00 µg/mL) and a negative control contained broth only. The minimal biofilm forming inhibition concentration (MBFIC) was calculated as linear regression of log_10_ (concentration of extracts) vs. % biofilm reduction. MBFIC_50_ and MBFIC_90_ values represent the lowest extract dilutions at which bacterial biofilm mass during formation was inhibited by 50% and 90%, compared to a negative control (inocula with broth) (GraphPad Prism version 6.00 for Windows; GraphPad Software, San Diego, CA, USA). All experiments were performed in triplicate and results are expressed as mean (Table 3).

### 3.9. Anti-Hemolytic Activity of Extracts

Experiments were performed on two strains, MSSA ATCC 6538 and MRSA MFBF 10679 using the Podvinje-M and Podvinje-W samples in sub-inhibitory concentrations (MIC/2 and MIC/4). The assay was performed according to Ferro et al., with certain alterations [38]. Blood samples were collected from one healthy volunteer, age 45, a non-smoker, after a written informed consent was obtained. The sample obtaining process was reviewed and approved by the Ethics Committee of Faculty of Pharmacy and Biochemistry, University of Zagreb Croatia (Document No. 643-02/18-01/02; 251-62-03-18-5 issued on 6 February 2018). A blood sample (2.5 mL) was collected using a heparinized tube and isolation of erythrocytes was performed immediately by centrifugation at 1500 rpm (10 min). Isolated erythrocytes were washed three times using PBS (pH 7.4) and their final concentration was 2%. Inocula with optical density of 0.5 McFarland units (~1.5 × 10^8^ CFU/mL) were freshly prepared in PBS (pH 7.4) from overnight bacterial cultures grown on MHA. Sample preparation was performed in brain heart infusion broth (BHI). Sample concentrations were as follows:*S**. aureus* ATCC 6538 → Podvinje-M: *c*_1_ (MIC/2) = 1.88 mg/mL and *c*_2_ (MIC/4) = 0.94 mg/mL; Podvinje-W *c*_1_ (MIC/2) = 4.17 mg/mL and *c*_2_ (MIC/4) = 2.08 mg/mLMRSA MFBF 10679 → Podvinje-M: *c*_1_ (MIC/2) = 1.25 mg/mL and *c*_2_ (MIC/4) = 0.63 mg/mL; Podvinje-W *c*_1_ (MIC/2) = 2.50 mg/mL and *c*_2_ (MIC/4) = 1.25 mg/mL.

Sample tubes with a final volume of 200 µL contained a mixture of extracts, 2% erythrocytes, and inoculum (10 µL of 0.5 McFarland units) and were incubated at 37 °C for 24 h, while rotating (250 rpm). A positive control contained Triton X-100 (2%). Negative controls contained vehicle (2% DMSO in PBS or pure PBS). After incubation, tubes were centrifuged (1500 rpm, 10 min) and the absorbance of 100 µL sample supernatants aliquots was measured at 540 nm in 96-well plates. Results were expressed as (1) a percentage (%) in relation to the hemolytic activity of each bacterial strain incubated with vehicle (negative control) (y_1_), and as a percentage (%) of hemolytic activity inhibition in relation to a negative control (y_2_) (Figure 4). The obtained results were also expressed as (2) hemoglobin equivalents calculated according to the calibration curve (*A* = 0.0773*c* + 0.0393) obtained as a relation of hemoglobin absorption (*A*; 540 nm) and its corresponding concentration (c; mg/mL) (Table 4).

### 3.10. TLC and Bioautography

#### 3.10.1. TLC

*E. cicutarium* extracts (Podvinje-M and Podvinje-W) were dissolved (c = 5 µg/µL) and applied on TLC silica gel F_254_ plates (20 × 20 cm; thickness 0.25 mm; Merck, Germany) in 8 mm wide bands using an automatic TLC sampler (ATS4, CAMAG, Muttenz, Switzerland). An acetonitrile/water/formic acid = 30:8:2 (V/V/V) mixture was used as a mobile phase. Plates developed for visualization were dried using a hair dryer, in a stream of cold air for 15 to 20 min. Visualization was performed using NSR (1% in methanol, m/V; Naturstoff reagent A; diphenylboric acid 2-aminoethyl ester; Sigma-Aldrich, St. Louis, MO 63118, USA) at 366 nm and 254 nm (Figure 5). Plates developed in parallel for bioautography under the same conditions were left to dry additionally overnight at room temperature in the dark.

#### 3.10.2. TLC-Bioautography

In order to perform antimicrobially directed bioautography assays of *E. cicutarium* extracts on *S. aureus*, the previously described methodology was used in a modified way [39]. A *S. aureus* ATCC 6538 overnight culture on TSA (aerobic incubation at 37 °C for 18 h in the dark; Sanyo MIR-533, Osaka, Japan) was used for inoculum preparation in PBS (pH 7.4). Freshly prepared inoculum was added to a molten Müller–Hinton agar (MHA) in 1:100 ratio, with a final concentration of bacterial cells ~1.5 × 10^6^ CFU/mL. Developed TLC chromatograms were put under UV light for 15 min to eliminate potential contamination. Inoculated MHA was then carefully applied on top of the TLC plates (2 mm thick), as an agar overlay variant of bioautography. Incubation was performed in a closed sterile plastic container for 24 h at 37 °C in the dark after agar solidification, with wet cotton balls to ensure humidity. Following incubation, bioautography plates were sprayed with 1% (m/V) solution of 2,3,5-triphenyltetrazolium chloride (Sigma-Aldrich, St. Louis, MO 63118, USA) in sterile physiological saline. Incubation was performed for 30 min at 37 °C in the dark. Inhibition zones were seen as clear spots around the active chromatogram zones with antibacterial activity against red background (Figure 5).

#### 3.10.3. Semi-Preparative TLC and UHPLC-LTQ OrbiTrap MS^4^

After detecting antimicrobially active zones on the developed TLC chromatograms, their *R*_F_ values were determined. Active bands were marked, scraped and semi-preparatively isolated. Extraction of bioactive compounds was performed using the previously described TLC mobile phase. In the following step, in order to remove the liquid phase, drying in a stream of nitrogen was applied. Composition of the obtained sample, i.e., the bioactive bands, was analyzed using UHPLC-LTQ OrbiTrap MS^4^ analysis, as described previously [14] (Table 5).

## 4. Conclusions

The obtained results confirm that *E. cicutarium*, which has a profiled and rich phytochemical composition, is a plant species with both an ethnopharmacological value and in vitro antimicrobial activity.

From the tested Gram-positive and Gram-negative in vitro bacterial models, and yeast, *E. cicutarium* showed the highest impact on the Gram-positive bacterial model *S. aureus*, including resistant clinical strains (MRSA).

Activity was confirmed for both types of extracts—water and methanol—in the agar well diffusion assay, the serial microdilution broth assay, and the “time-kill” assay, showing bactericidal and bacteriostatic activity, depending on experimental conditions. Biofilm formation inhibition and bacterial hemolysis was confirmed for the *S. aureus* model, as well. The obtained experimental data indicate that the observed antibacterial effects are not a result of bacterial cell wall disruption and, as such, open several new questions regarding the potential mechanism of antibacterial action.

For the first time, in this work we performed activity guided extracts’ fractionation using bioautography coupled with UHPLC-LTQ OrbiTrap MS^4^, after which identification of gallic acid derivatives and flavonol glycosides were identified as the most important compounds for the observed in vitro *S. aureus* antimicrobial activity.

Galloyl-shikimic acid stood out as the most abundant phytochemical in active fractions of both types of extracts, thereby motivating a further, more in-depth, investigation of its antimicrobial activity and synergy with other phytochemical compounds.

## Figures and Tables

**Figure 1 antibiotics-11-00492-f001:**
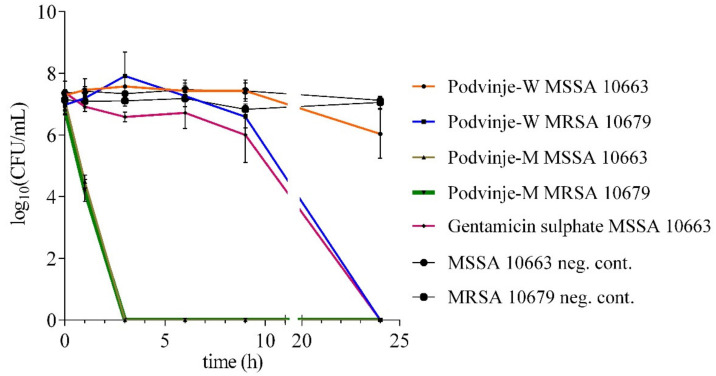
“Time-kill” assay of *E. cicutarium* water (Podvinje-W) and methanolic (Podvinje-M) extracts from the Podvinje locality on MSSA and MRSA strains (CFU, colony forming unit).

**Figure 2 antibiotics-11-00492-f002:**
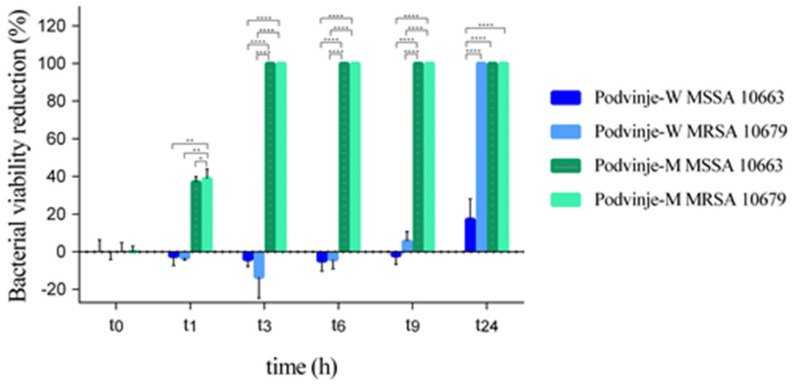
Bacterial viability reduction (%) in the “time-kill” assay for *E. cicutarium* water (Podvinje-W) and methanolic (Podvinje-M) extracts from the Podvinje locality on MSSA and MRSA strains at predefined time points t_0_, t_1_, t_3_, t_6_, t_9_ i t_24_ (one-way ANOVA and Tukey’s post hoc test; *, *p* ≤ 0.05; **, *p* ≤ 0.01; ****, *p* ≤ 0.0001).

**Figure 3 antibiotics-11-00492-f003:**
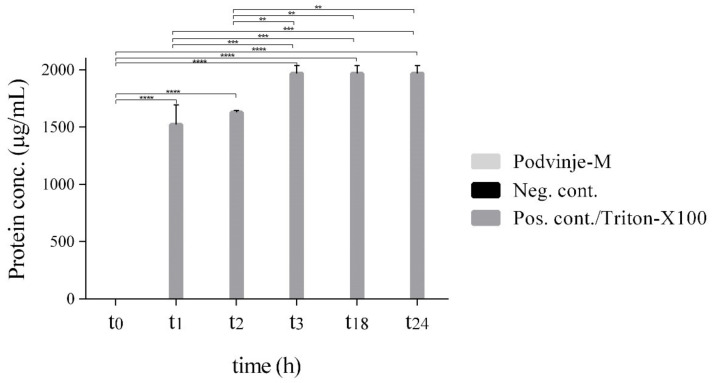
*S. aureus* ATCC 6538 protein leakage after cell integrity loss when treated with *E. cicutarium* methanolic extract from the Podvinje locality (Podvinje-M) and Triton-X100, compared to a negative control (N = 3; mean ± S.D; one-way ANOVA and Tukey’s post hoc test; **, *p* ≤ 0.01; ***, *p* ≤ 0.001; ****, *p* ≤ 0.000).

**Figure 4 antibiotics-11-00492-f004:**
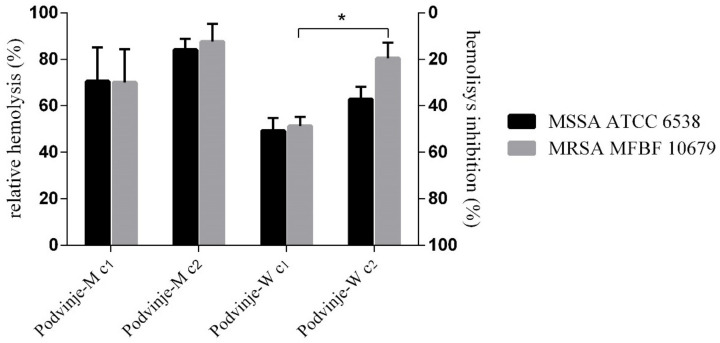
Anti-hemolytic activity of *E. cicutarium* water (Podvinje-W) and methanolic (Podvinje-M) extracts from the Podvinje locality on MSSA and MRSA strains (*, statistically significant difference among *c*_1_ and *c*_2_; *c*_1_ = MIC/2; *c*_2_ = MIC/4; *p* ≤ 0.01; *t*-test).

**Figure 5 antibiotics-11-00492-f005:**
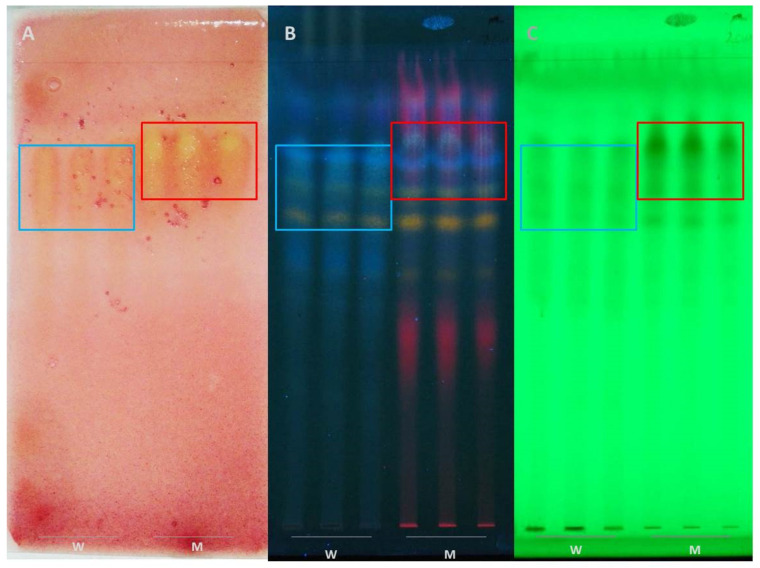
(**A**) Bioautography in situ—chromatograms of water (W) and methanolic (M) *E. cicutarium* extracts from the Podvinje locality (applied in triplicate) covered with inoculated MHA with *S. aureus* ATCC 6538 (1.5 × 10^6^ CFU/mL) and stained with bacterial viability indicator TTC (1%, m/V). Zones of bacterial growth inhibition are yellow, while the parts with viable bacteria are in red. (**B**) Chromatograms of water (W) and methanolic (M) *E. cicutarium* extracts of from the Podvinje locality (applied in triplicate) on TLC silica gel F_254_ plates, developed with mobile phase acetonitrile/water/formic acid = 30:8:2 (*v*/*v*/*v*), and visualized with NSR (1%, m/V) at 366 nm. (**C**) Same chromatograms at 254 nm.

**Table 1 antibiotics-11-00492-t001:** Susceptibility of *S. aureus* clinical isolates used in the study determined with VITEK^®^ 2 automated susceptibility testing system.

Strain	Antibiotic/MIC ^a^ (μg/mL)															
	Cefoxitin-Screen Test	Oxacillin	Gentamicin	Tobramycin	Levofloxacin	Moxyfloxacin	Erythromycin	Clindamycin	Linezolid	Teicoplanin	Vancomycin	Tetracycline	Tigecycline	Fosfomycin	Fusidic Acid	Trimethoprim/ Sulfamethox Azole
*S. aureus* MFBF ^b^ 10663 (MSSA ^c^)	-	2	≤0.5	≤1	1	0.5	≤0.25	≤0.25	2	2	2	≤1	≤0.12	16	≤0.5	≤10
*S. aureus* MFBF ^b^ 10679 (MRSA ^d^)	+	≥4	1	4	4	2	≥8	1	≥8	8	8	2	0.25	≥8	1	160

^a^ MIC—minimal inhibitory concentration; ^b^ Collection of Microorganisms of the Department of Microbiology, Faculty of Pharmacy and Biochemistry University of Zagreb; ^c^ MSSA—methicillin sensitive *S. aureus*; ^d^ MRSA—methicillin resistant *S. aureus*.

**Table 2 antibiotics-11-00492-t002:** In vitro antimicrobial activity screening of *E. cicutarium* water and methanolic extracts from four locations in Croatia (Podvinje, Plitvice, Trešnjevka, Buzin), for the agar well diffusion assay and the microdilution broth assay.

Sample		*S. aureus* ATCC 6538		MRSA ^f^ MFBF 10679		MSSA ^g^ MFBF 10663		*P. aeruginosa* ATCC 27853		*C. albicans* ATCC 90028	
		Mean ± S.D.									
		ZI ^a^ (mm)	MIC ^b^ (mg/mL)	ZI ^a^ (mm)	MIC ^b^ (mg/mL)	ZI ^a^ (mm)	MIC ^b^ (mg/mL)	ZI ^a^ (mm)	MIC ^b^ (mg/mL)	ZI ^a^ (mm)	MIC ^b^ (mg/mL)
		N = 5	N = 3	N = 5	N = 3	N = 5	N = 3	N = 5	N = 3	N = 5	N = 3
Podvinje	W ^d^	21 ± 1 ^1,2^	8.33 ± 2.9 ^2,3^	16 ± 1 ^2^	5.00 ± 0.0 ^1,2,3^	22 ± 2 ^3^	3.33 ± 1.4 ^1^	16 ± 1 ^1,2^	>20	6 ± 0	>20
	M ^e^	19 ± 2	3.75 ± 2.2	19 ± 2 ^2,3^	2.50 ± 0.0	22 ± 1 ^2,3^	3.33 ± 1.4	15 ± 2	20.00 ± 0.0 ^3^	15 ± 2 ^1^	>20
Plitvice	W ^d^	16 ± 2 ^2^	10.00 ± 0.0 ^2,3^	14 ± 2 *^,2^	10.00 ± 0.0 *^,3^	20 ± 2 ^3^	10.00 ± 0.0 *	13 ± 2	>20	6 ± 0	>20
	M ^e^	18 ± 1	10.00 ± 0.0	19 ± 1 ^3^	4.17 ± 1.4	22 ± 1 ^2,3^	3.33 ± 1.4	15 ± 1 ^3^	20.00 ± 0.0 ^3^	13 ± 1 ^2,3^	>20
Trešnjevka	W ^d^	12 ± 2 *^,3^	20.00 ± 0.0 *	6 ± 0 *^,3^	10.00 ± 0.0 *^,3^	16 ± 1 ^3^	6.67 ± 2.9	12 ± 1 *^,3^	>20	6 ± 0	>20
	M ^e^	17 ± 1	3.75 ± 2.2	16 ± 2	2.5 ± 0.0	20 ± 1	3.33 ± 1.4	15 ± 2	20.00 ± 0.0 ^3^	11 ± 1	>20
Buzin	W ^d^	20 ± 2	20.00 ± 0.0 *	16 ± 2	20.00 ± 0.0 *	20 ± 3 *	6.67 ± 2.9	15 ± 1	>20	6 ± 0	>20
	M ^e^	19 ± 2	6.67 ± 2.9	13 ± 2	6.67 ± 2.9	19 ± 1	3.33 ± 1.4	12 ± 2	10.00 ± 0.0	6 ± 0	>20
Gentamicin sulphate		17±1	0.001 ± 0.0	13 ± 2	0.001 ± 0.000	12 ± 1	0.001 ± 0.000	12 ± 1	0.003 ± 0.000	NT ^c^	NT ^c^
Nystatin		NT ^c^								25 ± 1	0.03 ± 0.00

^a^ ZI, zone of growth inhibition for c = 20 mg/mL; ^b^ MIC, minimal inhibitory concentration; ^c^ NT, not tested; ^d^ W, water extract; ^e^ M, methanolic extract; ^f^ MRSA, methicillin resistant *S. aureus*; ^g^ MSSA, methicillin sensitive *S. aureus*; * statistically significant difference compared to the methanolic extract from the same location (*p* < 0.05, *t*-test); ^1^ statistically significant difference compared to same type of extract from locality Plitvice (*p* < 0.05, one-way ANOVA and Tukey’s post hoc test); ^2^ statistically significant difference compared to the same extract type from the Trešnjevka locality (*p* < 0.05, one-way ANOVA and Tukey’s post hoc test); ^3^ statistically significant difference compared to same extract type from the Buzin locality (*p* < 0.05, one-way ANOVA and Tukey’s post hoc test); Remark: statistical analysis was conducted separately for each assay.

**Table 3 antibiotics-11-00492-t003:** Anti-biofilm activity of water (Podvinje-W) and methanolic (Podvinje-M) *E. cicutarium* extracts from the Podvinje locality on MSSA and MRSA (results shown as mean ± S.D; N = 3).

Sample or Control	MSSA ATCC 6538		MRSA MFBF 10679	
	MBFIC_50_ ^a^	MBFIC_90_ ^a^	MBFIC_50_ ^a^	MBFIC_90_ ^a^
	µg/mL			
Podvinje-W	1.05 ± 0.97	5095.37 ± 1143.88	61.84 ± 56.00	8509.34 ± 1268.83
Podvinje-M	0.41 ± 0.27	3083.98 ± 549.69	3.17 ± 5.30	8091.00 ± 477.24
Gentamicin sulphate	0.01 ± 0.01	0.99 ± 0.03	0.05 ± 0.03	1.66 ± 0.49

^a^ MBFIC = minimal biofilm-forming inhibition concentration, at which bacterial biofilm mass was inhibited by 50% (MBFIC_50_) and 90% (MBFIC_90_), compared to negative control.

**Table 4 antibiotics-11-00492-t004:** Anti-hemolytic activity of *E. cicutarium* water (Podvinje-W) and methanolic (Podvinje-M) extracts from the Podvinje locality on MSSA and MRSA strains, expressed as relative hemolysis (%) and bovine hemoglobin equivalents (A_540_ = 0.0773c + 0.0393; A_540_—absorbance at 540 nm; c—bovine hemoglobin concentration in mg/mL; R² = 0.9995).

Sample or Control	Hemoglobin Equivalents (mg/mL)		Relative Hemolysis (%)	
	MSSA ATCC 6538	MRSA MFBF 10679	MSSA ATCC 6538	MRSA MFBF 10679
Podvinje-M *c*_1_	5.36	5.33	70.50 ± 14.71	70.14 ± 14.22
Podvinje-M *c*_2_	6.56	7.41	84.15 ± 4.74	87.61 ± 7.57
Podvinje-W *c*_1_	3.47	3.90	49.30 ± 5.51	51.28 ± 4.00
Podvinje-W *c*_2_	4.55	6.60	62.90 ± 5.23	80.43 ± 6.66
NC (M)	7.53	7.97	100.00 ± 0.00	100.00 ± 0.00
PC (V)	7.73	8.42	100.00 ± 0.00	100.00 ± 0.00
PC/Triton X-100 (2%)	8.39		103.50	

NC—negative control; PC—positive control.

**Table 5 antibiotics-11-00492-t005:** Phenolic composition of antimicrobially active fractions of *E. cicutarium* water (Podvinje-W) and methanolic (Podvinje-M) extracts from the Podvinje locality, isolated semi-preparatively based on bioautography with MSSA ATCC 6538, and analyzed via UHPLC-LTQ OrbiTrap MS^4^ in negative ionization mode.

No	Compound Name	*t*_R_, min	Molecular Formula, [M–H]^-^	Calculated Mass, [M–H]^-^	Exact Mass, [M–H]^-^	Δ mDa	MS^2^ Fragments, (% Base Peak)	MS^3^ Fragments, (% Base Peak)	MS^4^ Fragments, (% Base Peak)	Podvinje-M	Podvinje-W
*Gallic acid derivatives*										mg/kg GAE *	
**1**	Galloyl hexoside isomer 1	2.84	**C_13_H_15_O_10_^-^**	331.06707	331.06312	3.95	125(12), 169(100), 170(8), 193(12), 211(28), 271(65), 272(9)	125(100)	81(33), 97(15), 107(100), 133(5)	35.43	15.41
**2**	Gallic acid	3.88	**C_7_H_5_O_5_^-^**	169.01425	169.01234	1.91	124(4), 125(100)	53(6), 79(17), 81(100), 83(3), 97(69), 107(16)	NA	140.51	54.64
**3**	Galloyl pentoside isomer 1	5.54	**C_12_H_13_O_9_^-^**	301.05651	301.05327	3.24	125(5), 149(55), 169(100), 170(7), 255(6), 256(5), 257(4)	125(100)	81(55), 97(20), 107(100)	3.87	9.69
**4**	Galloyl hexoside isomer 2	5.57	**C_13_H_15_O_10_^-^**	331.06707	331.06263	4.44	125(5), 169(100), 170(6), 223(3), 234(4), 285(3)	125(100)	81(61), 83(6), 97(100), 107(27), 239(16)	9.02	ND
**5**	Galloyl-shikimic acid	5.85	**C_14_H_13_O_9_^-^**	325.05651	325.05217	4.34	125(9), 169(100), 170(4)	125(100)	53(5), 79(3), 81(47), 97(53), 107(100)	375.42	531.45
**6**	Galloyl pentoside isomer 2	5.95	**C_12_H_13_O_9_^-^**	301.05651	301.05302	3.49	125(4), 149(58), 169(100), 170(4)	125(100)	81(32), 84(3), 97(34), 107(100), 109(3)	60.54	137.42
**7**	Protocatechuic acid	6.19	**C_7_H_5_O_4_^-^**	153.01933	153.01803	1.30	108(3), 109(100), 110(5)	65(100), 81(62)	NA	22.60	3.82
**8**	Methylgalloyl hexoside isomer 1	6.44	**C_14_H_17_O_10_^-^**	345.08272	345.07854	4.18	183(100), 184(4)	124(85), 168(100), 183(3)	124(100)	101.03	76.57
**9**	Digalloyl hexoside	6.50	**C_20_H_19_O_14_^-^**	483.07803	483.07317	4.86	169(12), 193(10), 211(14), 271(100), 272(11), 313(25), 331(22)	169(12), 211(100)	124(24), 165(10), 167(28), 168(100), 183(9)	11.46	7.53
**10**	Methylgalloyl hexoside isomer 2	6.81	**C_14_H_17_O_10_^-^**	345.08272	345.08100	1.72	183(100), 184(8), 299(12), 300(3), 323(3)	124(100), 168(93)	78(100), 96(25), 106(59)	15.05	ND
**11**	Corilagin	7.13	**C_27_H_21_O_18_^-^**	633.07334	633.06856	4.78	275(17), 301(100), 302(13), 419(5), 463(20), 613(10), 614(7)	185(34), 201(13), 229(61), 257(100), 284(24), 301(15)	185(100), 201(15), 213(6), 229(83), 230(6)	94.00	61.15
**12**	Methylgallate	7.29	**C_8_H_7_O_5_^-^**	183.02990	183.02832	1.58	124(100), 137(12), 153(12), 167(14), 168(100), 169(7), 183(9)	124(100)	78(100), 79(4), 106(45), 140(27)	24.67	4.10
**13**	Digalloyl-shikimic acid	7.31	**C_21_H_17_O_13_^-^**	477.06746	477.06271	4.75	169(25), 263(76), 289(100), 290(13), 307(31), 325(47), 453(12)	93(4), 137(100), 151(5), 245(9)	93(100)	14.97	12.28
**14**	Methylgalloyl-galloyl hexoside	7.38	**C_21_H_21_O_14_^-^**	497.09368	497.08902	4.66	183(6), 313(3), 345(100), 346(11), 465(11), 466(3)	183(100)	124(81), 168(100), 183(3)	16.34	19.42
**15**	Methylgalloyl-caffeoyl hexoside	8.59	**C_23_H_23_O_13_^-^**	507.11441	507.10987	4.54	179(5), 183(6), 323(18), 345(100), 346(13), 916(13), 917(6)	183(100)	124(90), 168(100)	5.63	2.33
**16**	Methylgalloyl-coumaroyl hexoside	9.21	**C_23_H_23_O_12_^-^**	491.11950	491.11539	4.11	183(14), 329(51), 330(10), 345(100), 346(13), 409(7), 457(9)	183(100)	124(71), 168(100), 183(3)	5.09	2.37
**17**	Trimethylellagic acid isomer 1	9.62	**C_17_H_11_O_4_^-^**	343.04594	343.04253	3.41	171(3), 297(4), 299(4), 315(3), 325(5), 328(100), 329(15)	313(100), 314(9)	285(41), 286(3), 298(100), 299(4)	78.57	61.98
**18**	Ellagic acid	9.65	**C_14_H_5_O_8_^-^**	300.99899	300.99735	1.64	185(41), 229(83), 255(48), 257(100), 271(61), 284(38), 301(37)	185(100), 186(12), 201(10), 213(18), 228(5), 229(70)	141(100), 157(46)	11.77	ND
**19**	Trimethylellagic acid isomer 2	10.74	**C_17_H_11_O_4_^-^**	343.04594	343.04226	3.68	295(4), 297(3), 325(3), 328(100), 329(17), 330(3)	313(100), 314(10)	285(40), 298(100), 299(8)	34.25	16.44
*Flavonol glycosides*										mg/kg RE *	
**20**	Quercetin 3-*O*-(2″-hexosyl)hexoside	7.50	**C_27_H_29_O_17_^-^**	625.14102	625.13645	4.57	271(18), 300(15), 300(100), 301(50), 445(26), 463(10), 505(11)	151(21), 179(25), 254(10), 255(53), 271(100), 272(21)	199(32), 215(28), 227(79), 243(100), 271(14)	2.57	112.39
**21**	Kaempferol 3-*O*-(2″-hexosyl)hexoside	7.81	**C_27_H_29_O_16_^-^**	609.14611	609.14166	4.45	255(11), 257(9), 284(47), 285(100), 286(11), 429(46), 430(8)	151(47), 213(31), 229(42), 241(50), 256(47), 257(100)	163(48), 187(13), 213(18), 229(100), 239(25)	2.77	11.46
**22**	Quercetin 3-*O*-(6″-rhamnosyl)glucoside (Rutin)	7.99	**C_27_H_29_O_16_^-^**	609.14611	609.14278	3.33	255(5), 271(7), 299(5), 300(42), 301(100), 302(13), 343(8)	151(78), 179(100), 256(10), 257(13), 272(13), 273(17)	151(100)	13.95	33.20
**23**	Quercetin 3-*O*-galactoside (Hyperoside)	8.31	**C_21_H_19_O_1__2_^-^**	463.08820	463.08349	4.71	300(12), 301(100), 302(11), 381(3), 445(4)	151(77), 179(100), 255(46), 257(12), 271(72), 272(23)	151(100)	18.32	13.40
**24**	Quercetin 3-*O*-hexuronide	8.31	**C_21_H_17_O_13_^-^**	477.06692	477.06196	4.96	301(100), 302(13), 315(7), 429(6), 431(3), 453(9)	107(5), 151(80), 179(100), 193(5), 257(13), 273(20)	151(100)	5.05	6.75
**25**	Kaempferol 7-*O*-(6″-rhamosyl)glucoside	8.41	**C_27_H_29_O_15_^-^**	593.15119	593.14656	4.63	257(3), 284(6), 285(100), 286(12)	197(20), 213(25), 229(49), 241(33), 257(100), 267(47)	163(75), 187(17), 213(32), 229(100), 239(29)	35.26	28.72
**26**	Isorhamnetin 3-*O*-(6″-rhamnosyl)glucoside (Narcissin)	8.48	**C_28_H_31_O_16_^-^**	623.16176	623.15692	4.84	255(3), 271(5), 300(14), 315(100), 316(13), 317(3)	272(6), 287(5), 300(100)	255(65), 271(100), 272(39)	97.90	107.05
**27**	Isorhamnetin 3-*O*-glucoside	8.82	**C_22_H_21_O_12_^-^**	477.10385	477.09994	3.91	271(8), 285(9), 314(100), 315(45), 316(6), 357(17), 453(7)	243(24), 271(74), 285(100), 286(50), 299(25), 300(44)	270(100)	11.10	6.40

* Compounds **1**–**19** were quantified as mg of gallic acid equivalents (GAE) per kg of fraction. Compounds **20**–**27** were quantified as mg of rutin equivalents (RE) per kg of fraction.

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
