# Peer review of "Antibacterial Fractions from *Erodium cicutarium* Exposed—Clinical Strains of *Staphylococcus aureus* in Focus"

_antibiotics, 2022, doi:10.3390/antibiotics11040492_

Round 1
Reviewer 1 Report
Comments to the Editor
The manuscript entitled “Antibacterial Fractions from Erodium cicutarium Exposed – Clinical Strains of Staphylococcus aureus In Focus” has been reviewed. The topic is quite innovative, interesting and scientifically well structured.
The MS is acceptable for publication. Only minor revisions are suggested.
Line 79 – The Table 1 is quite unclear in the section Strain, please
Line 110 – “S. aureus” needs Italic style
Line 115 – “P. aeuriginosa” needs Italic style
Line 116 – “E. cicutarium” needs Italic style
Line 163 – “Bactericial”, should be bactericidal
Author Response
The authors thank the Reviewer for suggested corrections and upgrades, all of them have been implemented as follows:
Line 79 – The Table 1 is quite unclear in the section Strain, please
Strains have been explained under Table 1, please see explenations under a, b, c and d.
Line 110 – “S. aureus” needs Italic style
Italic style has been applied.
Line 115 – “P. aeuriginosa” needs Italic style
Italic style has been applied.
Line 116 – “E. cicutarium” needs Italic style
Italic style has been applied.
Line 163 – “Bactericial”, should be bactericidal
Bactericial was corrected to bactericidal.
Reviewer 2 Report
Authors should reduce the number of keywords.
Line 41: A brief paragraph should be written in the introduction about the type of plant under study in terms of its spread, nature of growth, and its products
Line 79: The type of antibiotics used is not clear at the top of table 1.
Line 148: Photos for the agar plates of the MIC test should be added.
Line 345: I am asking if there is a relation between the geographical origin of the plant material and the extract activity of each? if so, a brief explanation of the effect of the geographical location on the plant material and so as to extracts should be added.
In my opinion, the phytochemical profiling of the different plant materials was very important to explain the difference in the bioactivity among different extracts.
Line 348: A brief description of the extraction method should be added.
Author Response
The authors thank the Reviewer for the very useful and interesting suggested corrections and upgrades.
Authors should reduce the number of keywords.
The number of keywords has been reduced to 8.
Line 41: A brief paragraph should be written in the introduction about the type of plant under study in terms of its spread, nature of growth, and its products
The paragraph has been added:
“Erodium cicutarium is a native species to the Mediterranean, but wide spread over the world, with fern-like, pinnate leaves forming a rosette, and small pink flowers, which after flowering transform to fruits with mericarps joined together in a spine-like style [3–6]. Literature data, as well as our previous research show the presence of a rich polyphenolic composition including tannins, flavonoids and hydroxycinnamic acids, as well as saponins, fatty acids, sugars and amino acids, along with a chemically diverse essential oil (hexadecanoic acid being the main compound) [6–14].”
Please note that, since data of such meaning has been described and published in our recent previous work, this description has initially been left out in order to prevent any form of potential self-plagiarism and as such should also be considered during this revision.
Ljoljić Bilić, V.; Stabentheiner, E.; Kremer, D.; Dunkić, V.; Jurišić Grubešić, R.; Vuković Rodríguez, J. Phytochemical and micromorphological characterization of Croatian populations of Erodium cicutarium. Nat. Prod. Commun. 2019, 14, 1–8, doi:10.1177/1934578X19856257.
Ljoljić Bilić, V.; Gašić, U.; Milojković-Opsenica, D.; Nemet, I.; Rončević, S.; Kosalec, I.; Vuković Rodriguez, J. First Extensive Polyphenolic Profile of Erodium cicutarium with Novel Insights to Elemental Composition and Antioxidant Activity. Chem. Biodivers. 2020, 17, doi:10.1002/cbdv.202000280.
Line 79: The type of antibiotics used is not clear at the top of table 1.
The antibiotics used in Table 1 are Cefoxitin, Oxacillin, Gentamicin,Tobramycin, Levofloxacin, Moxyfloxacin, Erythromycin, Clindamycin, Linezolid, Teicoplanin, Vancomycin, Tetracycline, Tigecycline, Fosfomycin, Fusidic acid and Trimethoprim/Sulfamethoxazole.
Line 148: Photos for the agar plates of the MIC test should be added.
Authors can proved some example photos of agar plates from the research by request of the reviewer, but the research was not design to be followed systematically by photographs of agar plates, neither are we aware that such kind of data presentation is common in this kind of methodology in more recent pubications.
Line 345: I am asking if there is a relation between the geographical origin of the plant material and the extract activity of each? if so, a brief explanation of the effect of the geographical location on the plant material and so as to extracts should be added.
In my opinion, the phytochemical profiling of the different plant materials was very important to explain the difference in the bioactivity among different extracts.
We thank the Reviewer for this very interesting question and comment. Plant samples in this reserach are from the Croatian area which is geographically relatively small and, as it was at least shown by obtained data from the described experimental conditions, the geographical changes are not in such proportions, that they would have impacted the phenolic composition in a significant amount. Also, if considering climate influence, the four localities have similar climate conditions. Ofcourse every locality has its own micro geographical and climate individuality, but as stated in our previous work, obtained experimental data showed that the phenolic composition was retively similar among localities. Only the methanolic extract from locality Trešnjevka showed slightly less phenolics to be present.
Ljoljić Bilić, V.; Gašić, U.; Milojković-Opsenica, D.; Nemet, I.; Rončević, S.; Kosalec, I.; Vuković Rodriguez, J. First Extensive Polyphenolic Profile of Erodium cicutarium with Novel Insights to Elemental Composition and Antioxidant Activity. Chem. Biodivers. 2020, 17, doi:10.1002/cbdv.202000280.
Line 348: A brief description of the extraction method should be added.
Description of extraction method has been added as follows:
“As described previously [14], the extract preparation included ultrasonication of 2.5 g of powdered herbal material with two types of solvents (12.5 mL) – methanol or water, at 45 °C for 45 min. After the first extraction and filtration, two re-extractions of the same plant material were made and the combined filtrates were evaporated and freeze-dried [14].”
Please note that, since data describing the extraction method were published in our recent previous work, this description has initially been left out in order to prevent any form of potential self-plagiarism and as such should also be considered during this revision.
Reviewer 3 Report
In this work, the authors conducted antibacterial studies on Erodium cicutarium. The results of performed in vitro screening on Staphylococcus aureus, Pseudomonas aeruginosa, and Candida albicans strains, show that E. cicutarium has antimicrobial activity, with a particular emphasis on clinical MSSA and MRSA. Observed antibacterial activity from preliminary screening results led to further activity-guided fractionation of water and methanol extracts using bioautography coupled with UHPLC-LTQ OrbiTrap MS4. It was determined that active fractions are predominantly formed by gallic acid derivatives and flavonol glycosides. Among the most active phytochemicals, galloyl-shikimic acid was identified as the most abundant compound. The work is interesting, but there are some issues that need to be revised before acceptance by Antibiotics.
- In table 1, there are some word that are not shown clearly, the authors should provide a complete Table1.
- In "Time-Kill" assay, why did the authors use Gentamycin sulfate as a positive control? The authors should have used the clinical drug vancomycin perhaps the results would have been a little better.
- Please provide a more accurate Figure 2. The significance analysis in the Figure 2 is not clear.
- Please provide a more beautiful Figure 3.
Author Response
The authors thank the Reviewer for the very useful and interesting suggested corrections and upgrades.
In table 1, there are some word that are not shown clearly, the authors should provide a complete Table1.
In Table 1, abbreviations MIC, MSSA, MRSA and MFBF have now been explained.
MIC – minimal inhibitory concentration; Collection of Microorganisms of the Department of Microbiology, Faculty of Pharmacy and Biochemistry University of Zagreb; MSSA – methicillin sensitive S. aureus; MRSA - methicillin resistant S. aureus.
In "Time-Kill" assay, why did the authors use Gentamycin sulfate as a positive control? The authors should have used the clinical drug vancomycin perhaps the results would have been a little better.
We thank the Reviewer for this usefull comment and agree, that vanomycin would have been a great option. However, in our work, as we started our study design using gentamycin sulfate as the positive control in the agar well assay and microdilution assay, we also decided to continue with gentamycin sulfate during the time-kill assay in order to keep the study overview uniform.
Please provide a more accurate Figure 2. The significance analysis in the Figure 2 is not clear.
A more accurate Figure 2 was provided. The aim of the Figure is to show a presentation of the data from the time-kill assay with the focus on the bacterial viability in every measured time point and the comparison among different types of extracts, as well as bacterial strains.
Please provide a more beautiful Figure 3.
Upgraded Figure 3 added in text.
Round 2
Reviewer 2 Report
The authors have carefully reviewed the manuscript. The manuscript is accepted for publication in its current form